# Geometry-Informed Neural Operator
# for Large-Scale 3D PDEs

**Zongyi Li**   **Nikola Borislavov Kovachki**   **Chris Choy**   **Boyi Li**   **Jean Kossaifi**

**Shourya Prakash Otta**   **Mohammad Amin Nabian**   **Maximilian Stadler**

**Christian Hundt**   **Kamyar Azizzadenesheli**   **Anima Anandkumar**

**NVIDIA**

## Abstract

We propose the geometry-informed neural operator (GINO), a highly efficient approach for learning the solution operator of large-scale partial differential equations with varying geometries. GINO uses a signed distance function (SDF) and point-cloud representations of the input shape and neural operators based on graph and Fourier architectures to learn the solution operator. The graph neural operator handles irregular grids and transforms them into and from regular latent grids on which Fourier neural operator can be efficiently applied. GINO is discretization-convergent, meaning the trained model can be applied to arbitrary discretizations of the continuous domain and it converges to the continuum operator as the discretization is refined. To empirically validate the performance of our method on large-scale simulation, we generate the industry-standard aerodynamics dataset of 3D vehicle geometries with Reynolds numbers as high as five million. For this large-scale 3D fluid simulation, numerical methods are expensive to compute surface pressure. We successfully trained GINO to predict the pressure on car surfaces using only five hundred data points. The cost-accuracy experiments show a $26,000\times$ speed-up compared to optimized GPU-based computational fluid dynamics (CFD) simulators on computing the drag coefficient. When tested on new combinations of geometries and boundary conditions (inlet velocities), GINO obtains a one-fourth reduction in error rate compared to deep neural network approaches.

## 1  Introduction

Computational sciences aim to understand natural phenomena and develop computational models to study the physical world around us. Many natural phenomena follow the first principles of physics and are often described as evolution on function spaces, governed by partial differential equations (PDE). Various numerical methods, including finite difference and finite element methods, have been developed as computational approaches for solving PDEs. However, these methods need to be run at very high resolutions to capture detailed physics, which are time-consuming and expensive, and often beyond the available computation capacity. For instance, in computational fluid dynamics (CFD), given a shape design, the goal is to solve the Navier-Stokes equation and estimate physical properties such as pressure and velocity. Finding the optimal shape design often requires solving thousands of trial shapes, each of which can take more than ten hours even with GPUs [1].

To overcome these computational challenges, recent works propose deep learning-based methods, particularly neural operators [2], to speed up the simulation and inverse design. Neural operators

37th Conference on Neural Information Processing Systems (NeurIPS 2023).

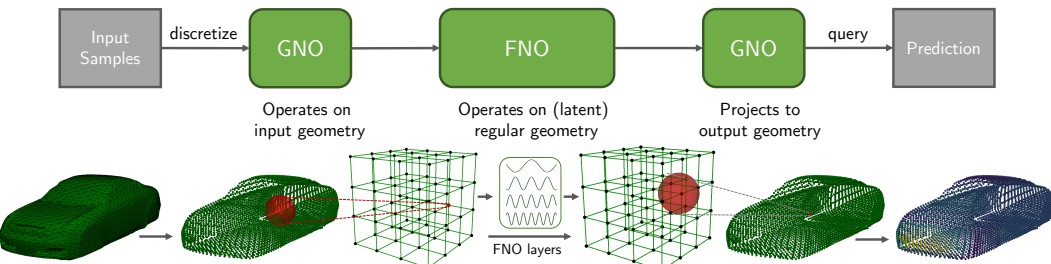

Figure 1: **The architecture of GINO**. The input geometries are irregular and change for each sample. These are discretized into point clouds and passed on to a GNO layer, which maps from the given geometry to a latent regular grid. The output of this GNO layer is concatenated with the SDF features and passed into an FNO model. The output from the FNO model is projected back onto the domain of the input geometry for each query point using another GNO layer. This is used to predict the target function (e.g., pressure), which is used to compute the loss that is optimized end-to-end for training.

generalize neural networks and learn operators, which are mappings between infinite-dimensional function spaces [2]. Neural operators are discretization convergent and can approximate general operators [3]. The input function to neural operators can be presented at any discretization, grid, resolution, or mesh, and the output function can be evaluated at any arbitrary point. Neural operators have shown promise in learning solution operators in partial differential equations (PDE) [3] with numerous applications in scientific computing, including weather forecasting [4], carbon dioxide storage and reservoir engineering [5], with a tremendous speedup over traditional methods. Prior works on neural operators developed a series of principled neural operator architectures to tackle a variety of scientific computing applications. Among the neural operators, graph neural operators (GNO) [2], and Fourier neural operators (FNO) [6] have been popular in various applications.

GNO implements kernel integration with graph structures and is applicable to complex geometries and irregular grids. The kernel integration in GNO shares similarities with the message-passing implementation of graph neural networks (GNN) [7], which is also used in scientific computing [8–10]. However, the main difference is that GNO defines the graph connection in a ball defined on the physical space, while GNN typically assumes a fixed set of neighbors, e.g., k-nearest neighbors, see Figure 5. Such nearest-neighbor connectivity in GNN violates discretization convergence, and it degenerates into a pointwise operator at high resolutions, leading to a poor approximation of the ground-truth operator using GNN. In contrast, GNO adapts the graph based on points within a physical space, allowing for universal approximation of operators. However, one limitation of graph-based methods is the computational complexity when applied to problems with long-range global interactions. To overcome this, prior works propose using multi-pole methods or multi-level graphs [11, 12] to help with global connectivity. However, they do not fully alleviate the problem since they require many such levels to capture global dependence, which still makes them expensive.

While GNO performs kernel integration in the physical space using graph operations, FNO leverages Fourier transform to represent the kernel integration in the spectral domain using Fourier modes. This architecture is applicable to general geometries and domains since the (continuous) Fourier transform can be defined on any domain. However, it becomes computationally efficient when applied to regular input grids since the continuous Fourier transform can then be efficiently approximated using discrete Fast Fourier transform (FFT) [13], giving FNO a significant quasi-linear computational complexity. However, FFT limits FNO to regular grids and cannot directly deal with complex geometries and irregular grids. A recent model, termed GeoFNO, learns a deformation from a given geometry to a latent regular grid [14] so that the FFT can be applied in the latent space. In order to transform the latent regular grid back to the irregular physical domain, discrete Fourier transform (DFT) on irregular grids is employed. However, DFT on irregular grids is more expensive than FFT, quadratic vs. quasi-linear, and does not approximate the Fourier transform in a discretization convergent manner. This is because, unlike in the regular setting, the points are not sampled at regular intervals, and therefore the integral does not take into account the underlying measure. Other attempts share a similar computational barrier as shown in Table 1, which we discussed in Section 5.

**In this paper**, we consider learning the solution operator for large-scale PDEs, in particular, 3D CFD simulations. We propose the geometry-informed neural operator (GINO), a neural operator

Table 1: **Computational complexity of standard deep learning models.** $N$ is the number of mesh points; $d$ is the dimension of the domain and degree is the maximum degree of the graph. Even though GNO and transformer both work on irregular grids and are discretization convergent, they become too expensive on large-scale problems.

| Model | Range | Complexity | Irregular grid | Discretization convergent |
|---|---|---|---|---|
| GNN | local | $O(N\text{degree})$ | ✔ | ✗ |
| CNN | local | $O(N)$ | ✗ | ✗ |
| UNet | global | $O(N)$ | ✗ | ✗ |
| Transformer | global | $O(N^2)$ | ✔ | ✔ |
| GNO (kernel) | radius $r$ | $O(N\text{degree})$ | ✔ | ✔ |
| FNO (FFT) | global | $O(N \log N)$ | ✗ | ✔ |
| GINO [**Ours**] | global | $O(N \log N + N\text{degree})$ | ✔ | ✔ |

architecture for arbitrary geometries and mesh discretizations. It uses a signed distance function (SDF) to represent the geometry and composes GNO and FNO architectures together in a principled manner to exploit the strengths of both frameworks.

The GNO by itself can handle irregular grids through graphs but is able to operate only locally under a limited computational budget, while the FNO can capture global interactions, but requires a regular grid. By using GNO to transform the irregular grid into a regular one for the FNO block, we can get the best of both worlds, i.e., computational efficiency and accuracy of the approximation. Thus, this architecture tackles the issue of expensive global integration operations that were unaddressed in prior works, while maintaining discretization convergence.

Specifically, GINO has three main components, $(i)$ **Geometry encoder**: multiple local kernel integration layers through GNO with graph operations, $(ii)$ **Global model**: a sequence of FNO layers for global kernel integration, and $(iii)$ **Geometry decoder**: the final kernel integral layers, as shown in Figure 1. The input to the GINO is the input surface (as a point cloud) along with the SDF, representing the distance of each 3D point to the surface. GINO is trained end-to-end to predict output (e.g., car surface pressure in our experiments), a function defined on the geometry surfaces.

**Geometry encoder:** the first component in the GINO architecture uses the surface (i.e., point cloud) and SDF features as inputs. The irregular grid representation of the surface is encoded through local kernel integration layers implemented with GNOs, consisting of local graphs that can handle different geometries and irregular grids. The encoded function is evaluated on a regular grid, which is concatenated with the SDF input evaluated on the same grid. **Global model:** the output of the first component is encoded on a regular grid, enabling efficient learning with an FNO using FFT. Our second component consists of multiple FNO layers for efficient global integration. In practice, we find that this step can be performed at a lower resolution without significantly impacting accuracy, giving a further computational advantage. **Geometry decoder:** the final component is composed of local GNO-based layers with graph operations, that decode the output of the FNO and project it back onto the desired geometry, making it possible to efficiently query the output function on irregular meshes. The GNO layers in our framework are accelerated using our GPU-based hash-table implementation of neighborhood search for graph connectivity of meshes.

We validate our findings on two large-scale 3D CFD datasets. We generate our own large-scale industry-standard Ahmed's body geometries using GPU-based OpenFOAM [15], composed of 500+ car geometries with $O(10^5)$ mesh points on the surface and $O(10^7)$ mesh points in space. Each simulation takes 7-19 hours on 16 CPU cores and 2 Nvidia V100 GPUs. Further, we also study a lower resolution dataset with more realistic car shapes, viz., Shape-Net car geometries generated by [16]. GINO takes the point clouds and SDF features as the input and predicts the pressure fields on the surfaces of the vehicles. We perform a full cost-accuracy trade-off analysis. The result shows GINO is $26,000\times$ faster at computing the drag coefficients over the GPU-based OpenFOAM solver, while achieving 8.31% (Ahmed-body) and 7.29% (Shape-Net car) error rates on the full pressure field. Further, GINO is capable of zero-shot super-resolution, training with only one-eighth of the mesh points, and having a good accuracy when evaluated on the full mesh that is not seen during training.

## 2 Problem setting

We are interested in learning the map from the geometry of a PDE to its solution. We will first give a general framework and then discuss the Navier-Stokes equation in CFD as an example. Let $D \subset \mathbb{R}^d$ be a Lipschitz domain and $\mathcal{A}$ a Banach space of real-valued functions on $D$. We consider the set of distance functions $\mathcal{T} \subset \mathcal{A}$ so that, for each function $T \in \mathcal{T}$, its zero set $S_T = \{x \in D : T(x) = 0\}$ defines a $(d-1)$-dimensional sub-manifold. We assume $S_T$ is simply connected, closed, smooth, and that there exists $\epsilon > 0$ such that $B_\epsilon(x) \cap \partial D = \varnothing$ for every $x \in S_T$ and $T \in \mathcal{T}$. We denote by $Q_T \subset D$, the open volume enclosed by the sub-manifold $S_T$ and assume that $Q_T$ is a Lipschitz domain with $\partial Q_T = S_T$. We define the Lipschitz domain $\Omega_T := D \setminus \bar{Q}_T$ so that, $\partial \Omega_T = \partial D \cup S_T$. Let $\mathcal{L}$ denote a partial differential operator and consider the problem

$$
\begin{aligned}
\mathcal{L}(u) &= f, &&\text{in } \Omega_T, \\
u &= g, &&\text{in } \partial \Omega_T,
\end{aligned} \tag{1}
$$

for some $f \in \mathcal{F}$, $g \in \mathcal{B}$ where $\mathcal{B}$, $\mathcal{F}$ denote Banach spaces of functions on $\mathbb{R}^d$ with the assumption that the evaluation functional is continuous in $\mathcal{B}$. We assume that $\mathcal{L}$ is such that, for any triplet $(T, f, g)$, the PDE (1) has a unique solution $u \in \mathcal{U}_T$ where $\mathcal{U}_T$ denotes a Banach space of functions on $\Omega_T$. Let $\mathcal{U}$ denote a Banach space of functions on $D$ and let $\{E_T : \mathcal{U}_T \to \mathcal{U} : T \in \mathcal{T}\}$ be a family of extension operators which are linear and bounded. We define the mapping from the distance function to the solution function

$$
\Psi : \mathcal{T} \times \mathcal{F} \times \mathcal{B} \to \mathcal{U} \tag{2}
$$

by $(T, f, g) \mapsto E_T(u)$ which is our operator of interest.

**Navier-Stokes Equation.** We illustrate the above abstract formulation with the following example. Let $D = (0, 1)^d$ be the unit cube and let $\mathcal{A} = C(\bar{D})$. We take $\mathcal{T} \subset \mathcal{A}$ to be some subset such that the zero level set of every element defines a $(d-1)$-dimensional closed surface which can be realized as the graph of a Lipschitz function and that there exists $\epsilon > 0$ such that each surface is at least distance $\epsilon$ away from the boundary of $D$. We now consider the steady Naiver-Stokes equations,

$$
\begin{aligned}
-\nu \Delta v + (v \cdot \nabla)v + \nabla p &= f, &&\text{in } \Omega_T, \\
\nabla \cdot v &= 0, &&\text{in } \Omega_T, \\
v &= q, &&\text{in } \partial D, \\
v &= 0, &&\text{in } S_T,
\end{aligned} \tag{3}
$$

where $v : \Omega_T \to \mathbb{R}^d$ is the velocity, $p : \Omega_T \to \mathbb{R}$ is the pressure, $\nu$ is the viscosity, and $f, q : \mathbb{R}^d \to \mathbb{R}^d$ are the forcing and boundary functions. The condition that $v = 0$ in $S_T$ is commonly known as a "no slip" boundary and is prevalent in many engineering applications. The function $q$, on the other hand, defines the inlet and outlet boundary conditions for the flow. We assume that $f \in H^{-1}(\mathbb{R}^d; \mathbb{R}^d)$ and $q \in C(\mathbb{R}^d; \mathbb{R}^d)$. We can then define our boundary function $g \in C(\mathbb{R}^d; \mathbb{R}^d)$ such that $g(x) = 0$ for any $x \in D$ with $\text{dist}(x, \partial D) \geq \epsilon$ and $g(x) = q(x)$ for any $x \in D$ with, $\text{dist}(x, \partial D) > \epsilon/2$ as well as any $x \notin D$. Continuity of $g$ can be ensured by an appropriate extension for any $x \in D$ such that $\text{dist}(x, \partial D) < \epsilon$ and $\text{dist}(x, \partial D) \geq \epsilon/2$ [17]. We define $u : \Omega_T \to \mathbb{R}^{d+1}$ by $u = (v, p)$ as the unique weak solution of (3) with $\mathcal{U}_T = H^1(\Omega_T; \mathbb{R}^d) \times L^2(\Omega_T)/\mathbb{R}$ [18]. We define $\mathcal{U} = H^1(D; \mathbb{R}^d) \times L^2(D)/\mathbb{R}$ and the family of extension operators $\{E_T : \mathcal{U}_T \to \mathcal{U}\}$ by $E_T(u) = \left(E_T^v(v), E_T^p(p)\right)$ where $E_T^v : H^1(\Omega_T; \mathbb{R}^d) \to H^1(D; \mathbb{R}^d)$ and $E_T^p : L^2(\Omega_T)/\mathbb{R} \to L^2(D)/\mathbb{R}$ are defined as the restriction onto $D$ of the extension operators defined in [19, Chapter VI, Theorem 5]. This establishes the existence of the operator $\Psi : \mathcal{T} \times H^{-1}(\mathbb{R}^d; \mathbb{R}^d) \times C(\mathbb{R}^d; \mathbb{R}^d) \to H^1(D; \mathbb{R}^d) \times L^2(D)/\mathbb{R}$ mapping the geometry, forcing, and boundary condition to the (extended) solution of the steady Navier-Stokes equation (3). Homomorphic extensions of deformation-based operators have been shown in [20]. We leave for future work studying the regularity properties of the presently defined operator.

## 3 Geometric-Informed Neural Operator

We propose a geometry-informed neural operator (GINO), a neural operator architecture for varying geometries and mesh regularities. GINO is a deep neural operator model consisting of three main components, $(i)$ multiple local kernel integration layers, $(ii)$ a sequence of FNO layers for global kernel integration which precedes $(iii)$ the final kernel integral layers. Each layer of GINO follows the form of generic kernel integral of the form (5). Local integration is computed using graphs, while global integration is done in Fourier space.

### 3.1 Neural operator

A neural operator $\Psi$ [3] maps the input functions $a = (T, f, g)$ to the solution function $u$. The neural operator $\Psi$ is composed of multiple layers of point-wise and integral operators,

$$\Psi = \mathcal{Q} \circ \mathcal{K}_L \circ \ldots \circ \mathcal{K}_1 \circ \mathcal{P}. \tag{4}$$

The first layer $\mathcal{P}$ is a pointwise operator parameterized by a neural network. It transforms the input function $a$ into a higher-dimensional latent space $\mathcal{P} : a \mapsto v_0$. Similarly, the last layer acts as a projection layer, which is a pointwise operator $\mathcal{Q} : v_l \mapsto u$, parameterized by a neural network $Q$. The model consists of $L$ layers of integral operators $\mathcal{K}_l : v_{l-1} \mapsto v_l$ in between.

$$v_l(x) = \int_D \kappa_l(x, y) v_{l-1}(y) \mathrm{d}y \tag{5}$$

where $\kappa_l$ is a learnable kernel function. Non-linear activation functions are incorporated between each layer.

### 3.2 Graph operator block

To efficiently compute the integral in equation (5), we truncate the integral to a local ball at $x$ with radius $r > 0$, as done in [2],

$$v_l(x) = \int_{B_r(x)} \kappa(x, y) v_{l-1}(y) \, \mathrm{d}y. \tag{6}$$

We discretize the space and use a Riemann sum to compute the integral. This process involves uniformly sampling the input mesh points and connecting them with a graph for efficient parallel computation. Specifically, for each point $x \in D$, we randomly sample points $\{y_1, \ldots, y_M\} \subset B_r(x)$ and approximate equation (6) as

$$v_l(x) \approx \sum_{i=1}^{M} \kappa(x, y_i) v_{l-1}(y_i) \mu(y_i), \tag{7}$$

where $\mu$ denotes the Riemannian sum weights corresponding to the ambient space of $B_r(x)$. For a fixed input mesh of $N$ points, the computational cost of equation (7) scales with the number of edges, denoted as $O(E) = O(MN)$. Here, the number of sampling points $M$ is the degree of the graph. It can be either fixed to a constant sampling size, or scale with the area of the ball.

**Encoder.** Given an input point cloud $\{x_1^{\mathrm{in}}, \ldots, x_N^{\mathrm{in}}\} \subset S_T$, we employ a GNO-encoder to transform it to a function on a uniform latent grid $\{x_1^{\mathrm{grid}}, \ldots, x_S^{\mathrm{grid}}\} \subset D$. The encoder is computed as discretization of an integral operator $v_0(x^{\mathrm{grid}}) \approx \sum_{i=1}^{M} \kappa(x^{\mathrm{grid}}, y_i^{\mathrm{in}}) \mu(y_i^{\mathrm{in}})$ over ball $B_{r_{\mathrm{in}}}(x^{\mathrm{grid}})$. To inform the grid density, GINO computes Riemannian sum weights $\mu(y_i^{\mathrm{in}})$. Further, we use Fourier features in the kernel [21]. For simple geometries, this encoder can be omitted, see Section 4.

**Decoder.** Similarly, given a function defined on the uniform latent grid $\{x_1^{\mathrm{grid}}, \ldots, x_S^{\mathrm{grid}}\} \subset D$, we use a GNO-decoder to query arbitrary output points $\{x_1^{\mathrm{out}}, \ldots, x_N^{\mathrm{out}}\} \subset \Omega_T$. The output is evaluated as $u(x^{\mathrm{out}}) \approx \sum_{i=1}^{M} \kappa(x^{\mathrm{out}}, y_i^{\mathrm{grid}}) v_l(y_i^{\mathrm{grid}}) \mu(y_i^{\mathrm{grid}})$ over ball $B_{r_{\mathrm{out}}}(x^{\mathrm{out}})$. Here, the Riemannian weight, $\mu(y_i^{\mathrm{grid}}) = 1/S$ since we choose the latent space to be regular grid. Since the queries are independent, we divide the output points into small batches and run them in parallel, which enables us to use much larger models by saving memory.

**Efficient graph construction.** The graph construction requires finding neighbors to each node that are within a certain radius. The simplest solution is to compute all possible distances between neighbors, which requires $O(N^2)$ computation and memory. However, as the $N$ gets larger, e.g., 10 $\sim$ 100 million, computation and memory become prohibitive even on modern GPUs. Instead, we use a hash grid-based implementation to efficiently prune candidates that are outside of a $\ell^\infty$-ball first and then compute the $\ell^2$ distance between only the candidates that survive. This reduces the computational complexity to $O(Ndr^3)$ where $d$ denotes unit density and $r$ is the radius. This can be efficiently done using first creating a hash table of voxels with size $r$. Then, for each node, we go over all immediate neighbors to the current voxel that the current node falls into and compute the distance between all points in these neighboring voxels. Specifically, we use the CUDA implementation from Open3D [22]. Then, using the neighbors, we compute the kernel integration using gather-scatter operations from torch-scatter [23]. Further, if the degree of the graph gets larger, we can add Nyström approximation by sampling nodes [2].

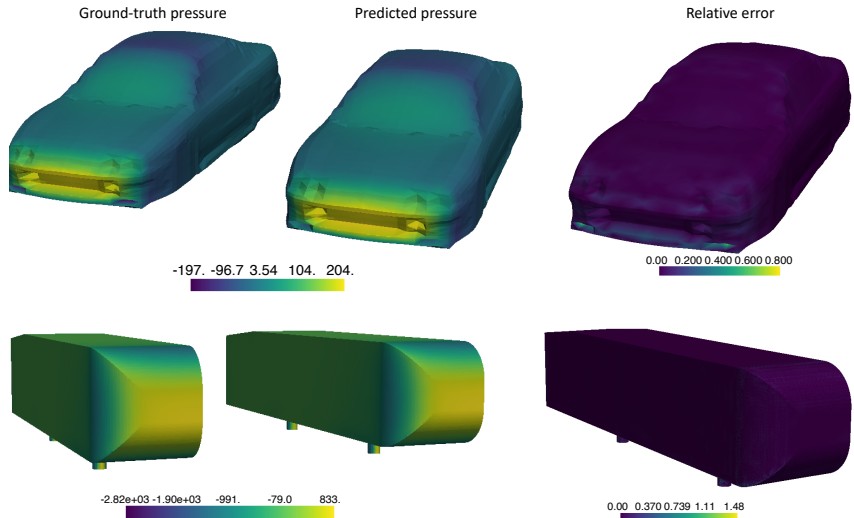

Figure 2: **Visualization of a ground-truth pressure and corresponding prediction by GINO** from the Shape-Net Car (**top**) and Ahmed-body (**bottom**) datasets, as well as the absolute error.

### 3.3 Fourier operator block

The geometry encoding $v_0$ and the geometry specifying map $T$, both evaluated on a regular grid discretizing $D$ are passed to a FNO block. We describe the basic FNO block as first outlined in [6]. We will first define global convolution in the Fourier space and use it to build the full FNO operator block. To that end, we will work on the $d$-dimensional unit torus $\mathbb{T}^d$. We define an integral operator with kernel $\kappa \in L^2(\mathbb{T}^d; \mathbb{R}^{n \times m})$ as the mapping $\mathcal{C} : L^2(\mathbb{T}^d; \mathbb{R}^m) \to L^2(\mathbb{T}^d; \mathbb{R}^n)$ given by

$$\mathcal{C}(v) = \mathcal{F}^{-1}\big(\mathcal{F}(\kappa) \cdot \mathcal{F}(v)\big), \qquad \forall\, v \in L^2(\mathbb{T}^d; \mathbb{R}^m)$$

Here $\mathcal{F}, \mathcal{F}^{-1}$ are the Fourier transform and its inverse respectively, defined for $L^2$ by the appropriate limiting procedure. The Fourier transform of the function $\kappa$ will be parameterized directly by some fixed number of Fourier modes, denoted $\alpha \in \mathbb{N}$. In particular, we assume

$$\kappa(x) = \sum_{\gamma \in I} c_\gamma e^{i\langle \gamma, x \rangle}, \qquad \forall\, x \in \mathbb{T}^d$$

for some index set $I \subset \mathbb{Z}^d$ with $|I| = \alpha$ and coefficients $c_\gamma \in \mathbb{C}^{n \times m}$. Then we may view $\mathcal{F} : L^2(\mathbb{T}^d; \mathbb{R}^{n \times m}) \to \ell^2(\mathbb{Z}^d; \mathbb{C}^{n \times m})$ so that $\mathcal{F}(\kappa)(\gamma) = c_\gamma$ if $\gamma \in I$ and $\mathcal{F}(\kappa)(\gamma) = 0$ if $\gamma \notin I$. We directly learn the coefficients $c_\gamma$ without ever having to evaluate $\kappa$ in physical space. We then define the full operator block $\mathcal{K} : L^2(\mathbb{T}^d; \mathbb{R}^m) \to L^2(\mathbb{T}^d; \mathbb{R}^n)$ by

$$\mathcal{K}(v)(x) = \sigma\big(Wv(x) + \mathcal{C}(v)\big), \qquad \forall\, x \in \mathbb{T}^d$$

where $\sigma$ is a pointwise non-linearity and $W \in \mathbb{R}^{n \times m}$ is a learnable matrix. We further modify the layer by learning the kernel coefficients in tensorized form, adding skip connections, normalization layers, and learnable activations as outlined in [24]. We refer the reader to this work for further details.

**Adaptive instance normalization.** For many engineering problems of interest, the boundary information is a fixed, scalar, inlet velocity specified on some portion of $\partial D$. In order to efficiently incorporate this scalar information into our architecture, we use a learnable adaptive instance normalization [25] combined with a Fourier feature embedding [21]. In particular, the scalar velocity is embedded into a vector with Fourier features. This vector then goes through a learnable MLP, which outputs the scale and shift parameters of an instance normalization layer [26]. In problems where the velocity information is not fixed, we replace the normalization layers of the FNO blocks with this adaptive normalization. We find this technique improves performance, since the magnitude of the output fields usually strongly depends on the magnitude of the inlet velocity.

Table 2: **Shape-Net Car dataset (3.7k mesh points).**

| Model | training error | test error |
|---|---|---|
| GNO | 18.16% | 18.77% |
| Geo-FNO (sphere) | 10.79% | 15.85% |
| UNet (interp) | 12.48% | 12.83% |
| FNO (interp) | 9.65% | 9.42% |
| GINO (encoder-decoder) | 7.95% | 9.47% |
| GINO (decoder) | 6.37% | **7.12%** |

We do a benchmark study with several standard machine-learning methods on the Shape-Net and Ahmed body datasets. The training error is normalized L2 error; the test error is de-normalized L2.

## 4    Experiments

We explore a range of models on two CFD datasets. The large-scale Ahmed-Body dataset, which we generated, and also the Shape-Net Car dataset from [16]. Both datasets contain simulations of the Reynold-Averaged Navier-Stokes (RANS) equations for a chosen turbulence model. The goal is to estimate the full pressure field given the shape of the vehicle as input. We consider GNO [2], MeshGraphNet [9], GeoFNO [14], 3D UNet [27] with linear interpolation, FNO [6], and GINO. We train each model for 100 epochs with Adam optimizer and step learning rate scheduler. The implementation details can be found in the Appendix. All models run on a single Nvidia V100 GPU.

### 4.1    Ahmed-Body dataset

We generate the industry-level vehicle aerodynamics simulation based on the Ahmed-body shapes [28]. The shapes are parameterized with six design parameters: length, width, height, ground clearance, slant angle, and fillet radius. We also vary the inlet velocity from 10m/s to 70m/s, leading to Reynolds numbers ranging from $4.35 \times 10^5$ to $6.82 \times 10^6$. We use the GPU-accelerated OpenFOAM solver for steady state simulation using the SST $k - \omega$ turbulence model [29] with 7.2 million mesh points in total with 100k mesh points on the surface. Each simulation takes 7-19 hours on 2 Nvidia v100 GPUs with 16 CPU cores. We generate 551 shapes in total and divide them into 500 for training and 51 for validation.

### 4.2    Shape-Net Car dataset

We also consider the Car dataset generated by [16]. The input shapes are from the ShapeNet Car category [30]. In [16], the shapes are manually modified to remove the side mirrors, spoilers, and tires. The RANS equations with the $k - \epsilon$ turbulence model and SUPG stabilization are simulated to obtain the time-averaged velocity and pressure fields using a finite element solver [31]. The inlet velocity is fixed at 20m/s (72km/h) and the estimated Reynolds number is $5 \times 10^6$. Each simulation takes approximately 50 minutes. The car surfaces are stored with 3.7k mesh points. We take the 611 water-tight shapes out of the 889 instances, and divide the 611 instances into 500 for training and 111 for validation.

As shown in Table 2 3 and Figure 2, GINO achieves the best error rate with a large margin compared with previous methods. On the Ahmed-body dataset, GINO achieves **8.31%** while the previous best method achieve 11.16%. On the Shape-Net Car, GINO achieves 7.12% error rate compared to 9.42% on FNO. It takes 0.1 seconds to evaluate, which is 100,000x faster than the GPU-parallel OpenFOAM solver that take 10 hours to generates the data. We further performance a full cost-accuracy analysis in the following section.

For ablations, we consider channel dimensions [32, 48, 64, 80], latent space [32, 48, 64, 80], and radius from 0.025 to 0.055 (with the domain size normalized to [-1, 1]). As depicted in Figure 4(a) and Table 4, larger latent spaces and radii yield superior performance.

### 4.3    Cost-accuracy analysis on drag coefficient

To compare the performance of our model against the industry-standard OpenFOAM solver, we perform a full cost-accuracy trade-off analysis on computing the drag coefficient, which is the standard design objective of vehicles and airfoils. The result shows GINO is 26,000x faster at computing the drag coefficients. Figure 3(b) below shows the cost-accuracy curve, measured in terms of inference

Table 3: **Ahmed-body dataset (100k mesh points).**

| Model | training error | test error |
|---|---|---|
| MeshGraphNet | 9.08% | 13.88% |
| UNet (interp) | 9.93% | 11.16% |
| FNO (interp) | 12.97% | 12.59% |
| GINO (encoder-decoder) | 9.36% | 9.01%% |
| GINO (decoder) | 9.34% | **8.31%** |

Previous works such as GNO and Geo-FNO cannot scale to large meshes with 100k points. We instead add the MeshGraphNet for graph comparison. Again, for UNet, FNO, and GINO, we fix the latent grid to $64 \times 64 \times 64$. The training error is normalized L2; the test error is de-normalized L2.

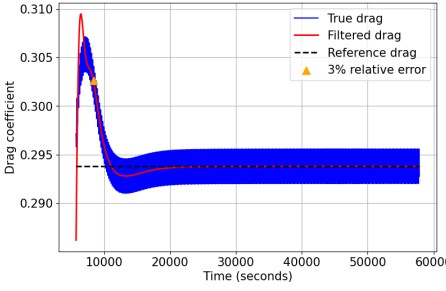 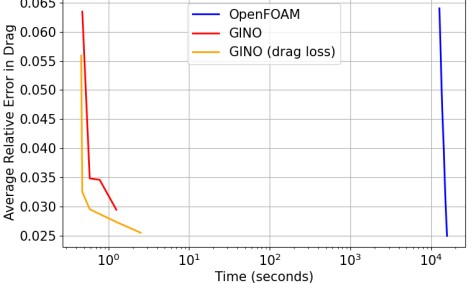

(a) Example drag coefficient computed after every iteration of the OpenFOAM solver. The reference drag corresponds to the last point of the filtered signal. Triangle indicates the the time when the solver reaches a $3\%$ relative error with respect to the reference drag.

(b) Cost-accuracy trade-off curve for OpenFOAM vs. GINO. Increasing cost of GINO is achieved by increasing the size of the latent space. Loss function for models on the red line includes only error in pressure and wall shear stress, while, for models on orange line, the loss also includes drag.

Figure 3: **Cost-accuracy trade-off analysis for the drag coefficient.**

time needed for a relative error in the drag coefficient for GINO and OpenFOAM. The detailed setup is discussed in the appendix.

## 4.4 Discretization-convergence and ablation studies

We investigate discretization-convergence by varying different parts of GINO. Specifically, we vary the latent grid resolution and the sampling rates for input-output meshes. In these experiments, we fixed the training and test samples to be the same, i.e., same latent grid resolution or sampling rate, but varied the shape and input conditions.

**Discretization-convergence wrt the latent grid.** Here, each model is trained and tested on (the same) latent resolutions, specifically 32, 48, 64, 80, and 88, and the architecture is the same. As depicted in Figure 4(a), GINO demonstrates a comparable error rate across all resolutions. A minor improvement in errors is observed when employing a larger latent space. Conversely, the errors associated with the UNet model grow as the resolution is decreased due to the decreasing receptive field of its local convolution kernels.

**Discretization-convergence in the input-output mesh.** Here, GINO is trained and tested with sub-sampled input-output meshes at various sampling rates (2x, 4x, 6x, 8x). As illustrated in Figure 4(b), GINO exhibits a consistent error rate across all sampling rates. A slight increase in errors is observed on coarser meshes.

**Zero-shot super-resolution.** GINO possesses the ability to perform zero-shot super-resolution. The model is trained on a coarse dataset, sub-sampled by 2x, 4x, 6x, and 8x, and subsequently tested on the full mesh, that is not seen during training. The error remains consistent across all sampling rates 4(c). This characteristic enables the model to be trained at a coarse resolution when the mesh is dense, consequently reducing the computational requirements.

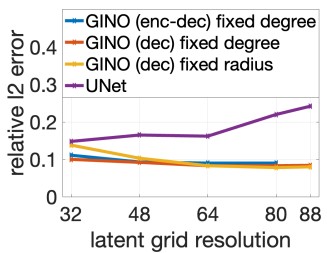
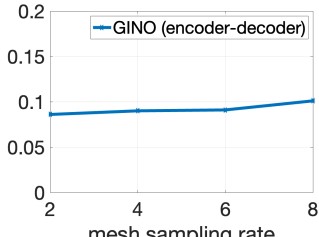
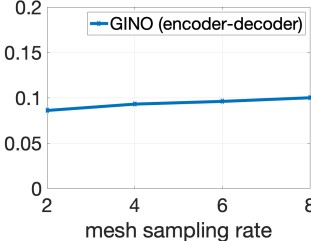

(a) Varying resolutions of the latent grid (same resolution for training and testing).

(b) Varying the sampling rates of the input-output mesh (same rate for training and testing).

(c) Train with a low sampling rate and test on full mesh (zero-shot super-resolution).

Figure 4: **Discretization-convergence studies and zero-shot super-resolution.**

## 5  Related Work

The study of neural operators and their extended applications in learning solution operators in PDE has been gaining momentum [3, 32–34]. A method that stands out is FNO, which uses Fourier transform [6]. The FNO and its variations have proven to highly accelerate the simulations for large-scale flow problems, including weather forecasting [4], seismology [35, 36] and multi-phase flow [5]. However, a challenge with the FNO is that its computation superiority is gained when applied on a regular grid, where the Fourier transform is approximated using FFT. Therefore, its reliance on FFT limits its use with irregular grids or complex geometries. There have been attempts to modify the FNO to work with these irregular structures, but scalability to large-scale 3D PDEs remains an issue. One such attempt is GeoFNO, which learns a coordinate transformation to map irregular inputs to a regular latent space [14]. This method, while innovative, requires a geometric discrete Fourier transform, which is computationally demanding and lacks discretization insurance. To circumvent this, GINO limits the Fourier transform to a local GNO to improve efficiency. The locality is defined assuming the metrics of the physical space.

Additionally, the Non-Equispaced Fourier neural solvers (NFS) merge the FNO with non-equispaced interpolation layers, a method similar to global GNO [37]. However, at the architecture level, their method replaces the integration of GNO with the summation of the nearest neighbor points on the graph. This step transitions this method to a neural network, failing to deliver a discretization convergent approach. The Domain-Agnostic Fourier Neural Operators (DAFNO) represents another attempt at improvement, applying an FNO to inputs where the geometry is represented as an indicator function [38]. However, this method lacks a strategy for handling irregular point clouds. Simultaneously, researchers are exploring the combination of FNO with the attention mechanisms [3] for irregular meshes. This includes the Operator Transformer (OFormer) [39], Mesh-Independent Neural Operator (MINO) [40], and the General Neural Operator Transformer (GNOT) [41]. Besides, the Clifford neural layers [42] use the Clifford algebra to compute multivectors, which provides Clifford-FNO implementations as an extension of FNO. The work [43] uses a physics-informed loss with U-Net for 3D channel flow simulation with several shapes. The work [44] innovatively proposes the use of multigrid training for neural networks that improves the convergence. Although these methods incorporate attention layers, which are special types of kernel integration [3] with quadratic complexity, they face challenges when scaling up for large-scale problems.

GNNs are incorporated in the prior attempts in physical simulations involving complex geometry, primarily due to the inherent flexibility of graph structures. Early research [7, 45–47] laid the foundation for GNNs, demonstrating that physical entities, when represented as graph nodes, and their interactions, as edges, could predict the dynamics of various systems. The introduction of graph element networks [48] marked a significant development, being the first to apply GNNs to PDEs by discretizing the domain into elements. The similar idea has been explored in term of continuous 3D point cloud convolution [49–51]. Another line of work, mesh graph networks [8–10], further explored PDEs in the context of fluid and solid mechanics. [52, 53] train a Graph convolutional neural works on the ShapeNet car dataset for inverse design. However, GNN architectures' limitations hinder their use in operator learning for PDEs. GNNs connect each node to its nearest neighbors according to the graph's metrics, not the metrics of the physical domain. As the input function's discretization becomes finer, each node's nearest neighbors eventually converge to the same node, contradicting the expectation of improved model performance with finer discretization. Furthermore, GNNs'

Input geometries                Discretized geometries at various fine to coarse levels of discretizations

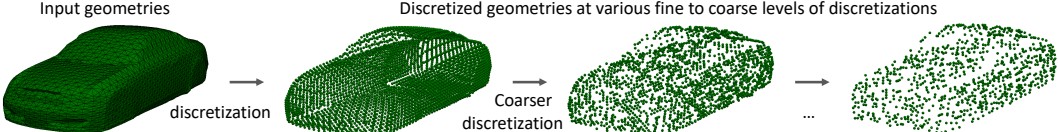

(a) **An input geometry** (continuous function) is first discretized into a series of points by subsampling it. Note that in practice, the discretization can be highly irregular. A key challenge with several scientific computing applications is that we want a method that can work on arbitrary geometries, but also that is discretization convergent, meaning that the method converges to a desired solution operator as we make the discretization finer.

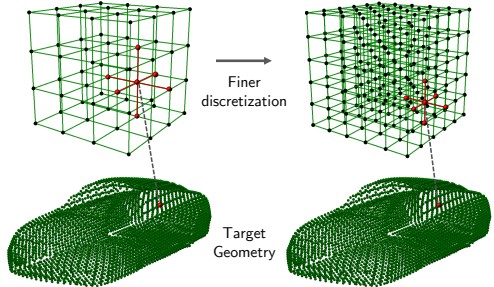 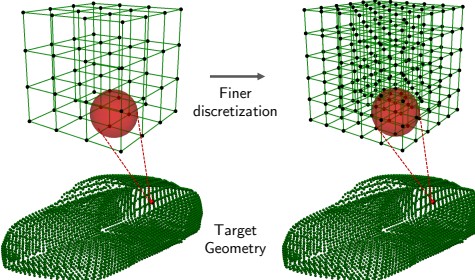

(b) **GNN** connects each point in the latent subspace (red) to its nearest neighbors in the original space (top). This is very discretization dependent, and as we increase the resolution (sample points more densely), the method becomes increasingly local and fails to capture context. In addition, the operator at the discretization limit is non-unique and depends on how the discretization is done.

(c) **GNO** instead connects each point in the latent subspace (red) to all its neighbors within an epsilon ball in the original space (top). This induces convergence to a continuum solution operator as we increase the resolution (sample points more densely). This means GNO converges to a unique operator as the discretization becomes finer and scales to large problems without becoming overly local.

Figure 5: **Comparison of GNN and GNO** as the discretization becomes finer. GNN is discretization dependent, while GNO is discretization convergent.

model behavior at the continuous function limit lacks a unique definition, failing the discretization convergence criterion. Consequently, as pointwise operators in function spaces at the continuous limit, GNNs struggle to approximate general operators between function spaces, Figure 5.

## 6   Conclusion

In this work, we propose the GINO model for 3D PDEs with complex geometries. The GINO model consists of the graph-kernel blocks for the encoder and decoder that go to a latent uniform space, where the Fourier blocks run on the latent space to capture the global interaction. We experiment on two CFD datasets: Shape-Net car geometries and large-scale Ahmed's body geometries, the latter encompassing over 600 car geometries featuring hundreds of thousands of mesh points. The evidence from these case studies illustrates that our method offers a substantial speed improvement, with a factor of 100,000 times acceleration in comparison to the GPU-based OpenFOAM solver. Concurrently, our approach has achieved one-fourth to one-half the error rates compared to prevailing neural networks such as 3D U-Net. This underscores the potential of our method to significantly enhance computational efficiency while maintaining a competitive level of accuracy within the realm of CFD applications. **Limitation:** The trained surrogate model is limited to a specific category of shapes. The quality of the model depends on the quality of the training dataset. For CFD with more complex shapes, it is not easy to obtain a large training dataset. We will explore physics-informed approaches [54] and generate time-dependent high-fidelity simulations in the future.

## Acknowledgments and Disclosure of Funding

ZL is supported by the Nvidia fellowship. NBK is grateful to the NVIDIA Corporation for support through full-time employment.

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

# 7 Appendix

## 7.1 Experiments and Ablations

Table 4: **Ablation on the Ahmed-body with different sizes of the latent space**

| Model | latent resolution | radius | training error | test error |
|-------|-------------------|--------|----------------|------------|
| GINO | 32 | 0.055 | 14.11% | 13.59% |
| GINO | 48 | 0.055 | 8.99% | 10.20% |
| GINO | 64 | 0.055 | 6.00% | 8.47% |
| GINO | 80 | 0.055 | 5.77% | **7.87%** |
| GINO | 32 | 0.110 | 8.66% | 10.10% |
| GINO | 48 | 0.073 | 7.25% | 9.17% |
| GINO | 64 | 0.055 | 6.00% | 8.47% |
| GINO | 80 | 0.044 | 6.22% | **7.89%** |

When fixing the radius, larger latent resolutions lead to better performance. The gaps become smaller when fixing the number of edges and scaling the radius correspondingly.

**Benchmarks.** This study analyzes several existing models, including GNO, GeoFNO, 3D UNet, and MeshGraphNet. All models are trained using the Adam optimizer for 100 epochs, with the learning rate halved at the 50th epoch. We consider starting learning rates such as [0.002, 0.001, 0.0005, 0.00025, 0.0001], with the most favorable results attained at the rates 0.00025 and 0.0001. For GeoFNO, a 2D spherical latent space is employed due to instability with 3D deformation, leading to a faster runtime than other 3D-based models. In the 3D UNet model, we evaluate channel dimensions ranging from [64, 128, 256] and depths from [4, 5, 6]. Utilizing a larger model can reduce UNet's error rate to 11.1%.

We added an experiment comparing GINO's performance with GNN. We used the MeshGraphNet [9], which is a common GNN method for physical simulations. The MeshGraphNet has three main parts: an encoder, a decoder, and a processor. Both the encoder and decoder use the same setup, with a channel size of 256. The processor has 15 layers of information passing, using edge and node blocks. It also has a size of 256. In total, the GNN model has about 10M parameters. The total number of edges is around 280k, which saturates a single NVIDIA V100 GPU with 32GB of memory. We set the learning rate to be 1e-4 with an exponential decay with a rate of 0.99985, similar to the original setup in [9].

For the GINO model, we consider channel dimensions [32, 48, 64, 80], latent space [32, 48, 64, 80], and radius from 0.025 to 0.055 (with the domain size normalized to [-1, 1]). As depicted in Figure 4(a) and Table 4, larger latent spaces and radii yield superior results.

Table 5: **Ablation on the Ahmed-body with different choices of the radius.**

| Model | radius 0.025 | | radius 0.035 | |
|-------|--------------|--|--------------|--|
| | training error | test error | training error | test error |
| GINO (encoder-decoder) | 12.91% | 13.07% | 8.65% | 10.32% |
| GINO (encoder-decoder, weighted) | 12.94% | 12.76% | 9.26% | 9.90% |
| GINO (decoder) | 12.62% | 12.74% | 8.82% | **9.39%** |

The choice of radius is significant. A larger radius leads to better performance for all models.

**Encoder and Decoder.** For the GINO model, we contemplate two configurations: an encoder-decoder design utilizing GNO layers for both input and output, and a decoder-only design which takes the fixed SDF input directly from the latent space and employs the GNO layer solely for output. When the input mesh significantly exceeds the latent grid in size, the encoder proves beneficial in information extraction. However, when the size of the latent grid matches or surpasses the input mesh, an encoder becomes redundant. As depicted in Table 5, both the encoder-decoder and decoder-only designs exhibit comparable performance.

**Parallelism.** Data parallelism is incorporated in the GNO decoder. In each batch, the model sub-samples 5,000 mesh points to calculate the pressure field. As query points are independent, they can be effortlessly batched. This parallel strategy allows for a larger radius in the decoder GNO. Without a parallel implementation, a radius of 0.025 leads to 300,000 edges, rapidly depleting GPU memory. Yet, with parallelism, the algorithm can handle a radius of 0.055. Implementing parallelism in the encoder is left for future exploration.

**Weights in the Riemann Sum.** As mentioned in the GNO section, the integral is approximated as a Riemann sum. In the decoder, the weight $\mu(y)$ is constant, reflecting the uniformity of the latent space. Conversely, in the encoder, weights are determined as the area of the triangle. For increased expressiveness, the weight is also integrated into the kernel, resulting in a kernel of the form $\kappa(x, y, \mu(y))$. However, it's worth noting that the encoder's significance diminishes when a large latent space is in use.

**Sub-sampling and Super-resolution.** The computational cost of the models increases rapidly with the number of mesh points. Training models with sub-sampled meshes saves significant computational resources. Discretization-convergent models can achieve such super-resolution; they can be trained on coarse mesh points and generalized to a fine evaluation mesh. We investigate the super-resolution capabilities of UNet (interp), FNO (interp), and GINO. As demonstrated in Table 6, GINO maintains consistency across all resolutions. UNet (interp) and FNO (interp) can also adapt to denser test meshes, albeit with a marginally higher error rate, based on linear interpolation. The results corroborate GINO's discretization-convergent nature.

**Hash-table-based Graph Construction.** For graph construction and kernel integration computation in this work, we utilize the CUDA implementation from Open3D [22] and torch-scatter [23], respectively. This approach is 40% faster compared to a previous GNO implementation that constructed the graphs with pairwise distance and used the PyTorch Geometric library[55]. These implementations are Incorporated into the GNO encoder and decoder in GINO.

In addition, the CUDA hash based implementation requires less memory footprint $O(Ndr^3)$ compare to the standard pairwise distance which requires $O(N^2)$ memory and computation complexity. For 10k points, hash-based implementation requires 6GB of GPU memory while the pairwise method requires 24GB of GPU memory; making the hash-based method more scalable for larger graphs.

Table 6: **Super-resolution on sub-sampled meshes**

| Model         Sampling rate | 1/2 | 1/4 | 1/6 | 1/8 |
|---|---|---|---|---|
| Unet (interp) | 16.5% | 13.8% | 13.9% | 15.6% |
| FNO (interp) | 14.2% | 14.1% | 13.3% | 11.5% |
| GINO (encoder-decoder) | 8.8% | 9.4% | 9.4% | 9.7% |

## 7.2 Drag Coefficient Comparison

For many engineering tasks, the goal is often to determine a single quantity of interest from a simulation which can then be used within an overall design process. In the design of automobiles, a sought after quantity is the drag coefficient of the vehicle. Intuitively, it is a number inversely proportional to the efficiency with which a vehicle passes through a fluid. Engineers are therefore often interested in designing geometries with minimal drag coefficients. For a fluid with unit density, the drag coefficient is defined as

$$c_d = \frac{2}{v^2 A} \left( \int_{\partial\Omega} p(x)\big(\hat{n}(x) \cdot \hat{i}(x)\big) \, dx + \int_{\partial\Omega} T_w(x) \cdot \hat{i}(x) \, dx \right) \tag{8}$$

where $\partial\Omega \subset \mathbb{R}^3$ is the surface of the car, $p : \mathbb{R}^3 \to \mathbb{R}$ is the pressure, $\hat{n} : \partial\Omega \to \mathbb{S}^2$ is the outward unit normal vector of the car surface, $\hat{i} : \mathbb{R}^3 \to \mathbb{S}^2$ is the unit direction of the inlet flow, $T_w : \partial\Omega \to \mathbb{R}^3$ is the wall shear stress on the surface of the car, $v \in \mathbb{R}$ is the speed of the inlet flow, and $A \in \mathbb{R}$ is the area of the smallest rectangle enclosing the front of the car.

For our Ahmed-body dataset, we train several GINO models (decoder) to predict the pressure on the surface of the car as well as the wall shear stress. Since the inlet flow is always parallel to the $x$-axis

Table 7: **Design of the Ahmed-body shapes**

| Parameters | steps | lower bound | upper bound |
|---|---|---|---|
| Length | 20 | 644 | 1444 |
| Width | 10 | 239 | 539 |
| Height | 5 | 208 | 368 |
| Ground Clearance | 2.5 | 30 | 90 |
| Slant Angle | 2.5 | 0 | 40 |
| Fillet Radius | 2.5 | 80 | 120 |
| Velocity | 4 | 10 | 70 |

i.e. $\hat{i}(x) = (-1, 0, 0)$, we predict only the first component of the wall shear stress. Since our results from Table 4 indicate that varying the size of the latent space has a significant effect on predictive performance, we make this the only hyper-parameter of the model and fix all others. We choose latent resolutions varying from 24 to 86. Furthermore, we consider two different loss functions. The first is simply the average of the relative $L^2$ errors for pressure and wall shear stress. The second includes in this average the relative error in the drag coefficient computed using equation (8). The drag is always computed from our full-field predictions and is never given as part of the output from the model.

To compare the performance of our model against the industry-standard OpenFOAM solver, we perform a full cost-accuracy trade-off analysis. During data generation, we keep track of the drag coefficient predicted by OpenFOAM after every iteration. While the coefficient converges with more iterations, this convergence is not monotone and can often appear quite noisy. This makes computing the error from the raw data not possible. We therefore apply a box filter to the raw signal to compute a filtered version of the drag which acts as smoother. We take as the reference drag, the drag at the last iteration of the filtered signal. To compute the number of iterations it takes for the solver to predict a drag coefficient at a given relative error, we trace back the predictions from the filtered signal and return the first time at which this prediction incurs the given error with respect to the reference drag. An example of this methodology is shown in Figure 3(a). The errors for our GINO model are computed with respect to the true drag coefficient from the last iteration of the solver. This is because we take as ground truth the pressure and wall shear stress from this last iteration and train our model to predict them.

Figure 3(b) shows the cost-accuracy curve, measured in terms of inference time needed for a relative error in the drag coefficient for GINO and OpenFOAM. The cost of GINO is computed as the time, averaged over the test set, needed to predict the drag coefficient by running the model. This time includes both data pre-processing (computing the SDF) as well as the model run-time and the drag calculation given the predicted fields. All models are ran on a single NVIDIA V100 GPU. The cost for OpenFOAM is computed as described in the previous paragraph and is averaged over the test set. The solver is ran on two NVIDIA V100 GPUs in parallel. We observe a four to five order of magnitude speed-up when using GINO. At a 3% relative error, we find the speed-up from our model which includes drag in the loss to be $26,000\times$. As we increase the size of the latent space, the cost of GINO grows, however, we observe a plateau in the drag error. This is common in machine learning models as the error from using finite data starts to dominate the approximation error. Furthermore, we use only the size of the latent space as a hyper-parameter, keeping the number of learnable parameters fixed. It is interesting to explore further how parametrically scaling the model impacts predictive power.

## 7.3 Data Generation

Industry-standard vehicle aerodynamics simulations are generated in this study, utilizing the Ahmed-body shapes as a foundation [28]. Examples are illustrated in Figure 6. These shapes are characterized by six design parameters: length, width, height, ground clearance, slant angle, and fillet radius, as outlined in Table 7. In addition to these design parameters, we include the inlet velocity to address a wide variation in Reynolds number. The inlet velocity varies from 10m/s to 70m/s, consequently resulting in Reynolds numbers ranging from $4.35 \times 10^5$ to $6.82 \times 10^6$. This varying input adds complexity to the problem. We identify the design points using the Latin hypercube sampling scheme for space filling design of experiments .

The simulations employ the GPU-accelerated OpenFOAM solver for steady-state analyses, applying the SST $k - \omega$ turbulence model. Consisting of 7.2 million mesh points in total, including 100k surface mesh points, each simulation is run on 2 NVIDIA V100 GPUs and 16 CPU cores, taking between 7 to 19 hours to complete.

For this study, the focus is solely on the prediction of the pressure field. It is our hope that this dataset can be utilized in future research, potentially aiding in full-field simulation of the velocity field, as well as the inverse design.

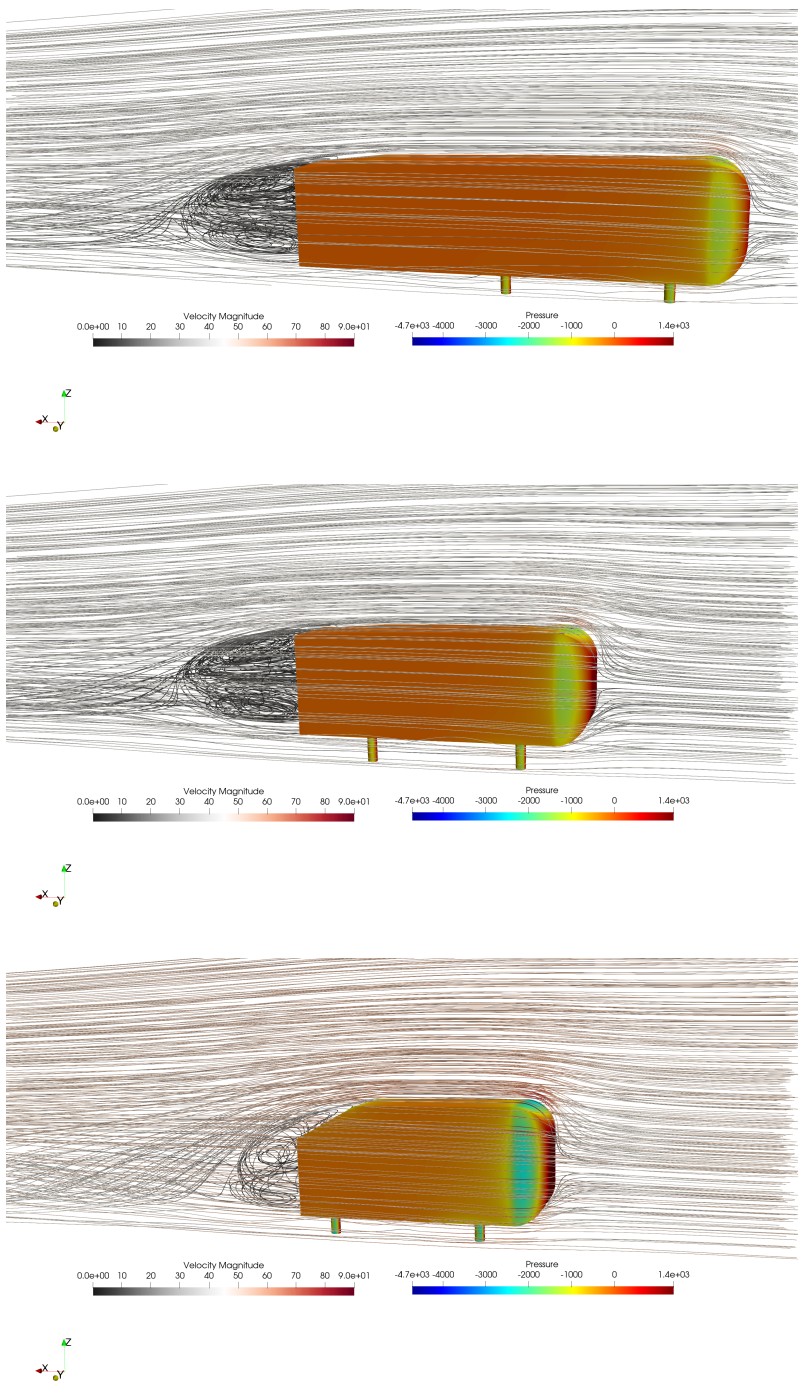

Figure 6: Illustrations of the Ahmed-body dataset

