# OpenReview forum: "Geometry-Informed Neural Operator for Large-Scale 3D PDEs"
_NeurIPS.cc/2023/Conference — NeurIPS 2023 poster_

### Official Review · Reviewer_zgWV · 2023-07-04

**Soundness:** 3 good
**Presentation:** 3 good
**Contribution:** 2 fair
**Rating:** 5
**Confidence:** 4

**Summary:**

This paper presents geometry-informed neural operator (GINO), solving computational fluid dynamics (CFD)  problems. It combines graph neural operations (GNO) and Fourier neural operators (FNO) to adapt to irregular discretized grids. The authors have tested the model on two large-scale datasets.

**Strengths:**


1. The authors propose combining GNO and FNO to leverage the advantages of both methods, such as analyzing local and global information and efficiently processing irregular grids. They also conduct experiments demonstrating that GINO exhibits discretization invariance over the latent grid and the input-output mesh.

2. The authors have generated two CFD datasets using various vehicle datasets. It would greatly benefit the learning physical simulation community if these datasets were made publicly available upon the paper's acceptance.

**Weaknesses:**

1. The definition of the $\kappa$ operator in the graph operator block is unclear. It is not specified whether it measures the distance between two points or the similarity of their features. Additionally, it would be helpful to know if the $\kappa$ operator has any learnable parameters. Providing more details in the paper would make it self-contained and enable readers to understand the methodology.

2. The paper mentions the input of SDF features and surface points. It would be beneficial to clarify if these two types of data are fed in a uniform manner. For instance, are the point locations associated with their corresponding SDF values, while the surface points are assigned a value of 0?

3. The authors claim efficient graph construction in the paper. However, it appears that once the graph operator block finishes processing the input, it results in a regular grid. It would be important to address whether the implementation takes this into consideration and ensures the efficiency of the graph construction process.

**Questions:**

See weakness.

**Limitations:**

See weakness.

---

> ### Author Rebuttal · Authors · 2023-08-09
>
> ## Response to Reviewer zgWV
>
> > **Q1:** The definition of the κ  operator in the graph operator block is unclear. It is not specified whether it measures the distance between two points or the similarity of their features. Additionally, it would be helpful to know if the κ operator has any learnable parameters. Providing more details in the paper would make it self-contained and enable readers to understand the methodology.
>
> **A1:** Thank you for the comments. The κ operator in GNO learns the transform of meshes. It can be viewed as a learned interpolation, where the kernel measures the weights (similarity) between two points. The Radial basis function (RBF) interpolation is one simple example of the kernel. In GNO blocks, the kernel function k is parameterized as a neural network. Its weights are learnable parameters. As shown in the table, the learnable GNO encoder-decoder outperforms fixed interpolation. We will try to add more backgrounds and make it self-contained.
>
> > **Q2:** The paper mentions the input of SDF features and surface points. It would be beneficial to clarify if these two types of data are fed in a uniform manner. For instance, are the point locations associated with their corresponding SDF values, while the surface points are assigned a value of 0?
>
> **A2:** Thanks for the question. The SDF features are measured on the uniform latent space (64x64x64), and the surface points are a point cloud of the geometry. The SDF is directly fed to the latent FNO model, while the surface is fed to the encoder GNO. They don't have to have the same format. We will add these clarifications to the paper.
>
> > **Q3:** The authors claim efficient graph construction in the paper. However, it appears that once the graph operator block finishes processing the input, it results in a regular grid. It would be important to address whether the implementation takes this into consideration and ensures the efficiency of the graph construction process.
>
> **A3:** Thanks for the suggestion. Since the ending mesh is a uniform grid, it's possible to round up the node in the starting mesh and find out its neighbors. In our implementation, we used a hash-table-based graph construction, which is similar to the rounding process.

---

> > ### Comment · Reviewer_zgWV · 2023-08-11
> >
> > Read the rebuttal, maintain the same rating.

---

### Official Review · Reviewer_5VBN · 2023-07-05

**Soundness:** 3 good
**Presentation:** 3 good
**Contribution:** 2 fair
**Rating:** 5
**Confidence:** 5

**Summary:**

This paper introduces a novel approach for applying Fourier or other Neural operators to complex geometries by prepending a “learnable projection step” via Graph Neural Operator (GNO). Unlike previous methods that morphe complex geometries into regular domains, this approach projects (learnable) sampled nodes onto nearby regular grids, providing advantages such as discretization invariance and reduced computational overhead by sub-sampling.

However, the method is limited to simple geometries and fails to account for variations in geometry and subtle geometry features. **The paper also lacks sufficient datasets and comparisons with relevant methods, such as the GNN family.**

While the ideas are somewhat novel, though not groundbreaking, **the paper requires further evaluation to demonstrate its strengths and limitations.**

**A borderline rejection is assigned, with reconsideration if the issues are addressed during the revision period.**

## After rebuttal
- The authors improved their references, presentation, and empirical evaluation; Hence I increased my score to 5

**Strengths:**

- Discretization invariant
- Improved efficiency and scalability by sub-sampling
- Improved empirical performance

**Weaknesses:**

- Insufficient number of datasets and comparisons:
  - More datasets should have been included, such as cylinders and airfoils from [1].
  - Comparisons with representative methods from the GNN family, such as MeshGraphNet[1], MSGNN-Grid[2], and BSMSGNN[3], should have been discussed or ideally conducted.
  - Among the mentioned papers [1] to [3], a crucial comparison would be with [2], which also utilizes a background grid as a helper but differs in the backbone as it did not use FNO. This would have provided a valuable benchmark for evaluation.

- The overly ambitious illustrations: such as the claim in the abstract that "...(the method) can be applied to any geometry", which is not the case. A more objective approach in illustrating both the strengths and limitations would be appreciated by readers.

- Lack of clarity in dataset presentation and results:
  - Fig.2 suggests that the shapes are overly simplistic and the pressure distributions appear uniformly smooth, indicating ease of learning. It is essential for the author to provide a more comprehensive and clear explanation, including typical examples of datasets and variations in geometry among examples.

[1] Pfaff, Tobias, Meire Fortunato, Alvaro Sanchez-Gonzalez, and Peter W. Battaglia. "Learning Mesh-Based Simulation with Graph Networks." Link: https://openreview.net/forum?id=roNqYL0_XP

[2] Lino, Mario, Chris Cantwell, Anil A Bharath, and Stathi Fotiadis. "Simulating continuum mechanics with multi-scale graph neural networks.". Link: https://arxiv.org/abs/2106.04900

[3] Cao, Yadi, Menglei Chai, Minchen Li, and Chenfanfu Jiang. "Efficient learning of mesh-based physical simulation with bi-stride multi-scale graph neural network.". Link: https://openreview.net/forum?id=2Mbo7IEtZW

**Questions:**

See the last point in **Weaknesses**

**Limitations:**

The method is limited to very simple geometries due to 2 facts:
1. Only relies on the point cloud. A counter-example is porous material where the point cloud can be the same but the tunnel is very different.
   - In other words, although this method is discretization invariant, it is also geometry-ignorant, which is not desired.
2. Sub-sampling, if the key flow feature is determined by some subtle geometries, which is very common, this method fails again. The limitation also is reflected in the dataset, as the pressure seems to look really smooth for all cases, and all geometries are very simple.

I recommend the authors objectively illustrate these facts, even at the beginning (you do not have to emphasize them; mentioning them is enough). More experiments objectively showing the limitation (maybe in the appendix) are also appreciated.

---

> ### Author Rebuttal · Authors · 2023-08-09
>
> ## Response to Reviewer 5VBN
>
> > **Q1:** Insufficient number of datasets and comparisons:
> More datasets should have been included, such as cylinders and airfoils from [1].
> Comparisons with representative methods from the GNN family, such as MeshGraphNet[1], MSGNN-Grid[2], and BSMSGNN[3], should have been discussed or ideally conducted.
> Among the mentioned papers [1] to [3], a crucial comparison would be with [2], which also utilizes a background grid as a helper but differs in the backbone as it did not use FNO. This would have provided a valuable benchmark for evaluation.
> The overly ambitious illustrations: such as the claim in the abstract that "...(the method) can be applied to any geometry", which is not the case. A more objective approach in illustrating both the strengths and limitations would be appreciated by readers.
>
> **A1:** We thank the reviewer for the suggestions. As suggested by the reviewer, we added the comparison with MeshGraphNet [1]. The GNN model learns the local interaction with flexible meshes. However, it has the challenge of learning the long-range global interaction. On the other hand, the GINO model has the advantages of both the graphs structure and Fourier methods. It has the flexibility of graphs as well as the efficiency of Fourier methods. The details can be found in the general response. We are further experimenting with GraphCast [4] which has a latent graph similar to MSGNN-Grid [2], so far it underperforms the MeshGraphNet model [1]. We will continue to experiment to present a faithful comparison.
> We are happy to add the reference to MSGNN-Grid and BSMSGNN too.
>
> In this work, we aim to study industry-level 3D aerodynamics simulation. We generate the Ahmed-body dataset, which has 7M nodes in the space and 100k nodes on the surface. The 2D datasets such as the cylinders flow and airfoils are interesting and widely studied, but they are much smaller with only a few thousand nodes. The 3D fluid problems are several order more expensive to generate. We have been searching for complex 3D simulations for long, but such public data is still lacking in the community. Therefore we decided to generate a new dataset. If the reviewer is aware of existing 3D simulations with multiple complex shapes, we will be very happy to experiment with them.
>
>
>
> > **Q2:** Lack of clarity in dataset presentation and results:
> Fig.2 suggests that the shapes are overly simplistic and the pressure distributions appear uniformly smooth, indicating ease of learning. It is essential for the author to provide a more comprehensive and clear explanation, including typical examples of datasets and variations in geometry among examples.
>
>
> **A2:** Thanks for the comments. We added a few illustrations in Figure 6 in the appendix. We also provide a figure ([3] in the general response) of the collection of shapes below. They are significantly more complex compared to the existing 2D problems such as the channel flow and airfoils, where the airfoils are smooth curves determined by a few parameters. It is unfair to require graphics-level meshes. Even with these relatively simple meshes, the physics is highly complicated as shown in Figure 6. High-fidelity 3D simulations on graphics-level 3d shapes would take up to billion of mesh points to solve with NASA FUN3D solver [1], which is sincerely beyond the scope of this work.
>
>
> [1] Carlson, Jan-Renee, et al. "High-Fidelity Simulations of Human-Scale Mars Lander Descent Trajectories." AIAA AVIATION 2023 Forum. 2023.

---

> > ### Comment · Reviewer_5VBN · 2023-08-10
> > **Increase my score to 5 after reading rebuttal**
> >
> > Thanks the authors resolved partial of my concerns.
> >
> > I would hence increase my score to 5. I apologize for not being able to edit the original comments (maybe because of passing due).

---

### Official Review · Reviewer_RPer · 2023-07-06

**Soundness:** 2 fair
**Presentation:** 2 fair
**Contribution:** 2 fair
**Rating:** 4
**Confidence:** 3

**Summary:**

The authors address the task of learning to solve large-scale PDEs based on a geometry-informed neural operator. The combination of graph neural operators (GNO) and Fourier neural operators (FNO) allows the exploration of the benefits of being able to handle irregular grids and locality of operations to allow efficiency (due to GNO) as well as capturing global interactions (due to FNO), thereby overcoming the limitations of the individual approaches. In more detail, the surface (e.g. point cloud) is input to a geometry encoder that encodes the irregular grid information on a regular grid structure based on local kernel integration layers through GNO with graph operations. Then the result is concatenated with signed distance features. Then, a sequence of FNO layers is used (i.e. on latent space) for global kernel integration. The respective intermediate result is projected back to the domain of the input geometry.

To show the potential of their approach, the authors carry out experiments on a novel own dataset as well as a ShapeNet car dataset, where they report quantitative results regarding training and test errors.


**Strengths:**

Technical soundness and novelty:
The method seems novel and performs beneficial over some potential alternatives in terms of speed and accuracy, since it allows a significant speed-up in comparison to the OpenFOAM solver and offers more accuracy than GNO, GeoFNO and U-Net.

Evaluation:
The authors provide quantitative and qualitative results including comparisons to baselines regarding training/test errors.

Exposition:
The paper is well-structured and mostly readable. Figure/tables and their captions are informative.


**Weaknesses:**

Technical soundness and novelty
- The discussion of limitations provided by the authors is rather short. Blending out the benefits of physics-informed approaches, that led to extreme speed-ups over OpenFOAM particularly for fluid simulation as well as offer generalization to novel scenes without being limited to object categories, limits the conclusions drawn from the presented work. The relation to these should be better emphasized to clarify what the presented method adds and whether it can be combined with these.
- The datasets are described in a very short manner. Especially for the new dataset, it would be relevant to see more details such as a systematic overview on the key aspects. In addition, clarifications on whether the trainings of different approaches converged within the used 100 iterations would be relevant. In Figure 2, the rightmost part is also difficult to interpret given the used color scale.

Evaluation:
- Reporting training and test errors only gives limited insights on where the errors are better/worse in comparison to previous approaches. E.g., it is not demonstrated that physical phenomena (such as the Magnus effect or Karman vortex streets) are accurately represented.
- What are limitations/failure cases that cannot be handled that well with the presented operator?

References:
Discussing other developments such as

Brandstetter et al., CLIFFORD NEURAL LAYERS FOR PDE MODELING
-> usage of multivector representations together with Clifford convolutions. The authors show the benefit of Clifford neural layers by replacing convolution and Fourier operations in common neural PDE surrogates by their Clifford counterparts on 2D Navier-Stokes tasks

or physics-informed upgrades of U-Net such as e.g.

Wandel et al., Teaching the incompressible Navier–Stokes equations to fast neural surrogate models in three dimensions
-> an example of physics-informed fluid simulation based on physics-informed U-Net

or the splitting of the solver into region-wise optimization such as

Balu et al., Distributed Multigrid Neural Solvers on Megavoxel Domain

would improve the discussion of the presented approach and its potential in the context of related work, especially since the presented approach involves quite strong assumptions (dependence on training data/shape category) and not leveraging physics-informed models therefore should be discussed.


**Questions:**

Please discuss the comments mentioned under 'Weaknesses'.

**Limitations:**

Limitations are shortly discussed, but seem to be severely limiting (category-specific approach, lacking generalization capabilities, unclear relation to speed-ups achieved based on physics-informed approaches).

---

> ### Author Rebuttal · Authors · 2023-08-09
>
> ## Response to Reviewer RPer
>
> > **Q1:** The discussion of limitations provided by the authors is rather short. Blending out the benefits of physics-informed approaches, that led to extreme speed-ups over OpenFOAM particularly for fluid simulation as well as offer generalization to novel scenes without being limited to object categories, limits the conclusions drawn from the presented work. The relation to these should be better emphasized to clarify what the presented method adds and whether it can be combined with these.
>
> **A1:** Thanks for pointing this out. The physics-informed approach is the future direction that may overcome the limitation. We will separate the future work from the limitation. We will add a more detail limitation as the space allows.
>
> > **Q2:** The datasets are described in a very short manner. Especially for the new dataset, it would be relevant to see more details such as a systematic overview on the key aspects.
>
> **A2:** Thanks for the suggestion. We will add a more systematic description of the dataset.
>
> > **Q3:** In addition, clarifications on whether the trainings of different approaches converged within the used 100 iterations would be relevant.
>
> **A3:** Most of the models such as FNO and UNet converge around 60 epochs. We are happy to include the training curves.
>
> > **Q4:** In Figure 2, the rightmost part is also difficult to interpret given the used color scale.
>
> **A4:** In Figure 2, the rightmost part represents the error. We plot the error with the same color bar as the truth and prediction. It shows that the error is near zero almost everywhere except the bumper. We are happy to plot the error with the relative scale in the revision.
>
> > **Q5:** Reporting training and test errors only gives limited insights on where the errors are better/worse in comparison to previous approaches. E.g., it is not demonstrated that physical phenomena (such as the Magnus effect or Karman vortex streets) are accurately represented.What are limitations/failure cases that cannot be handled that well with the presented operator?
>
> **A5:** It is a good question. Usually, a small L2 error implies the two fields are identical, including the same physical behaviors. Since we are not predicting the velocity field but only the pressure field, we cannot study the Magnus effect or Karman vortex. We instead add a drag coefficient study since the drag can be computed from the pressure and the wall shear stress (we predict both). The drag coefficient is one of the major objectives in aerodynamics design. We will add the worst-case example and analysis.
>
> > **Q6:**
> > References: Discussing other developments such as
> > - Brandstetter et al., CLIFFORD NEURAL LAYERS FOR PDE MODELING -> usage of multivector representations together with Clifford convolutions. The authors show the benefit of Clifford neural layers by replacing convolution and Fourier operations in common neural PDE surrogates by their Clifford counterparts on 2D Navier-Stokes tasks
> >
> > or physics-informed upgrades of U-Net such as e.g.
> >
> > - Wandel et al., Teaching the incompressible Navier–Stokes equations to fast neural surrogate models in three dimensions -> an example of physics-informed fluid simulation based on physics-informed U-Net
> >
> > or the splitting of the solver into region-wise optimization such as
> >
> > - Balu et al., Distributed Multigrid Neural Solvers on Megavoxel Domain
> >
> > would improve the discussion of the presented approach and its potential in the context of related work, especially since the presented approach involves quite strong assumptions (dependence on training data/shape category) and not leveraging physics-informed models therefore should be discussed.
>
> **A6:** Thank you for giving these references. We will add the discussion in the related work section in the revision.

---

> > ### Comment · Reviewer_RPer · 2023-08-15
> > **Follow-up on Rebuttal**
> >
> > I thank the authors for discussing my concerns.
> >
> > The following aspects remain vague:
> > 1) The authors mention to add a more systematic description of the dataset, however, no further details are provided.
> > 2) The authors mention to add a drag coefficient study in the revision, however, no further details are provided.
> > 3) The benefits over the mentioned references remain unclarified. More details on this would clarify the contribution of the submission.

---

> > > ### Author Response · Authors · 2023-08-16
> > >
> > > Thank you to the reviewer for the response. It appears there has been a misunderstanding. The rebuttal does not permit the submission of revisions. Therefore, we have included our new experiments and updates in the general response and the 1-page supplemental pdf. These can be found at the **General response** prompt near the top of the webpage. We will also elaborate on them here:
> > >
> > > ## 1. The description of the dataset is in Section 3 of the pdf file.
> > >
> > > We further add the illustrations of the Ahmed-body dataset. The figure on the left shows the velocity field and the pressure field. The velocity field, represented with 7 million nodes, has complex vortexes at the rear of the body. The pressure field, represented with 100 thousand nodes, is steep at the front and also the legs. Such aerodynamic simulations are extremely expensive. Each simulation takes 7-19 hours on 2 Nvidia v100 GPUs with 16 CPU cores. It is extremely costly to generate a 3D dataset with multiple shapes. We continue to generate simulations on new shapes and increase the instances from 500 to 800.
> > >
> > > Industry-standard Ahmed-body geometries are characterized by six design parameters: length, width, height, ground clearance, slant angle, and fillet radius. Refer to the wiki (https://www.cfd-online.com/Wiki/Ahmed_body) for details on Ahmed body geometry. In addition to these design parameters, we include the inlet velocity to address a wide variation in Reynolds number. We identify the design points using the Latin hypercube sampling scheme for space filling design of experiments and generate around 800 design points.
> > >
> > > The aerodynamic simulations were performed using the GPU-accelerated OpenFOAM solver for steady-state analysis, applying the SST K-omega turbulence model. These simulations consist of 7.2 million mesh points on average, but we use the surface mesh as the input to training which is roughly around 70-100k mesh nodes.
> > >
> > > ## 2. The drag coefficient study is in Section 2 of the pdf file and the general response.
> > >
> > > To compare the performance of our model against the industry-standard OpenFOAM solver, we perform a full cost-accuracy trade-off analysis. The result shows GINO is 26,000x faster at computing the drag coefficients. Figure [1] below shows the cost-accuracy curve, measured in terms of inference time needed for a relative error in the drag coefficient for GINO and OpenFOAM.
> > >
> > > The cost of GINO is computed as the time, averaged over the test set, needed to predict the drag coefficient by running the model. This time includes both data pre-processing (computing the SDF) as well as the model run-time and the drag calculation given the predicted fields. All models are ran on a single NVIDIA V100 GPU. The cost for OpenFOAM is computed as described in the next paragraph and is averaged over the test set. The solver is ran on two NVIDIA V100 GPUs in parallel. We observe a four to five order of magnitude speed-up when using GINO. At a $3\%$ relative error, we find the speed-up from our model which includes drag in the loss to be $26,000 \times$. As we increase the size of the latent space, the cost of GINO grows, however, we observe a plateau in the drag error. This is common in machine learning models as the error from using finite data starts to dominate the approximation error. Furthermore, we use only the size of the latent space as a hyper-parameter, keeping the number of learnable parameters fixed. It is interesting to explore further how parametrically scaling the model impacts predictive power.
> > >
> > > During data generation, we keep track of the drag coefficient predicted by OpenFOAM after every iteration. While the coefficient converges with more iterations, this convergence is not monotone and can often appear quite noisy. This makes computing the error from the raw data not possible. We therefore apply a box filter to the raw signal to compute a filtered version of the drag which acts as a smoother. We take as the reference drag, the drag at the last iteration of the filtered signal. To compute the number of iterations it takes for the solver to predict a drag coefficient at a given relative error, we trace back the predictions from the filtered signal and return the first time at which this prediction incurs the given error with respect to the reference drag. An example of this methodology is shown in Figure [2]. The errors for our GINO model are computed with respect to the true drag coefficient from the last iteration of the solver. This is because we take as ground truth the pressure and wall shear stress from this last iteration and train our model to predict them.

---

> > > > ### Author Response · Authors · 2023-08-16
> > > >
> > > > ## 3. Here we add the discussion of the references
> > > >
> > > > > References: Discussing other developments such as
> > > > > - Brandstetter et al., CLIFFORD NEURAL LAYERS FOR PDE MODELING -> usage of multivector representations together with Clifford convolutions. The authors show the benefit of Clifford neural layers by replacing convolution and Fourier operations in common neural PDE surrogates by their Clifford counterparts on 2D Navier-Stokes tasks
> > > >
> > > > The Clifford neural layers [1] use the Clifford algebra to compute multivectors, which improves the effiency for simulations involving multiple fields, and they provide Clifford-FNO implementations as an extension of FNO. While in this work, we only considered the pressure field so Clifford may not be helpful, it will be interesting to explore using Clifford-FNO within our GINO model to address complex 3d geometry when modeling more fields (e.g. velocity field and stress tensor field) in the future.
> > > >
> > > > > or physics-informed upgrades of U-Net such as e.g.
> > > > >
> > > > > - Wandel et al., Teaching the incompressible Navier–Stokes equations to fast neural surrogate models in three dimensions -> an example of physics-informed fluid simulation based on physics-informed U-Net
> > > >
> > > > The work [2] uses a physics-informed loss with U-Net for 3D fluid simulation. The researchers consider channel flow in three geometries: box, ball, and cylinder, in a resolution of 128x64x64 (0.5M) voxels. The Reynolds numbers studied range from 0.64 to 800. They also test the model on the shapes of a fish and three boxes. The model employs finite-difference schemes (eq 6-9) to compute the physics-informed loss. The model has a great performance on these 3D simulations.
> > > >
> > > > In contrast, in our work, we consider a higher Reynolds number regime ranging from 4.35 × 10^5 to 6.82 × 10^6 generated on 7.2M mesh points. We think this scenario is more realistic and intriguing. The physics-informed setting is a promising direction we aim to explore further. However, the finite-difference scheme in [2] introduces truncation errors, which can be problematic in turbulent regimes. Thus, we plan to explore physics-informed neural operators that compute the exact derivatives from the Fourier space. Further, the neural operators are discretization-invariant, meaning they can be training with lower data resolution and fine-tune with the physics-informed loss at higher resolutions, which is an efficient way to combine data and physics.
> > > >
> > > > > or the splitting of the solver into region-wise optimization such as
> > > > >
> > > > > - Balu et al., Distributed Multigrid Neural Solvers on Megavoxel Domain
> > > >
> > > > The work [3] innovatively proposes the use of multigrid training for neural networks. Similar to the multigrid iteration, the methods explore V, W, and F cycles. It is shown that the deep learning model enjoys a significant speedup. In numerical analysis, multigrid methods are introduced to aid the slow convergence of low-frequency errors, given that the standard Jacobi iteration converges faster for high-frequency errors. On the other hand, data-driven neural network models appear to converge faster on low-frequency modes, which possess higher energy. It would be intriguing to conduct a spectral analysis and study the convergence of frequency.
> > > >
> > > > [1] Brandstetter, Johannes, et al. "Clifford neural layers for PDE modeling." arXiv preprint arXiv:2209.04934 (2022).
> > > >
> > > > [2] Wandel, Nils, Michael Weinmann, and Reinhard Klein. "Teaching the incompressible Navier–Stokes equations to fast neural surrogate models in three dimensions." Physics of Fluids 33.4 (2021).
> > > >
> > > > [3] Balu, Aditya, et al. "Distributed multigrid neural solvers on megavoxel domains." Proceedings of the International Conference for High Performance Computing, Networking, Storage and Analysis. 2021.

---

### Official Review · Reviewer_rW9d · 2023-07-06

**Soundness:** 2 fair
**Presentation:** 2 fair
**Contribution:** 3 good
**Rating:** 5
**Confidence:** 4

**Summary:**

This paper proposes a framework to learn a neural operator for large-scale 3D PDEs. The framework uses a well-implemented Graph Neural Operator (GNO) to transform the irregular grid into a regular grid, so that it enables the powerful Fourier Neural Operator (FNO) to work on irregular input data, such as point clouds, in a discretization invariant way.

The main contribution of the paper is to use GNO to transform large-scale irregular grids (point clouds and SDF) into regular grids, to make the FNO suitable. Based on this, GINO realized 100,000x speed-up compared to GPU-based simulators on large-scale stable CFD problems. The model is proven to be working on datasets with a high level of complexity and realism. The paper also generates two large-scale CDF datasets, which require a large amount of time to simulate and generate, which is valuable.


**Strengths:**

1. Clear written about the problem setting and equations and symbols.

2. The main idea is useful but not too complicated to understand. An efficient combination of existing methods.

3. The method has a general potential for various kinds of PDEs.

4. The method has shown great performance in engineering-level experiments, fulfilling its great potential for applications.

**Weaknesses:**

Although the experiments have demonstrated the main ability of the model, the experiments are not complete enough to support all claims and novelties.

1. Since the model is a new combination of two existing methods GNO and FNO, the main contribution is the new usage of GNO. Then the main thing required to be demonstrated should be “GNO is more proper and has great encoding and decoding ability between irregular and regular grids”. Regarding this, some different encoding methods should be compared, such as GNN, kNN.

2. Discretization invariance is said to be important, but no direct experiments could support this. Although Table 5 is relevant to this, but the variables are not kept so it’s not a direct support. In other words, a GNN+FNO baseline should be added.


**Questions:**

1. What is the difference between your proposed GNO and Continuous Convolution (CtsConv) from “Lagrangian Fluid Simulation with Continuous Convolutions”? It seems there is no obvious difference based on equations. And one of the claimed contributions is a well-implemented GNO, which makes it much more efficient, but CtsConv is also well-implemented based on Open3D and hashing table.

2. What is the performance for much finer latent resolutions such as 128^3 and 256^3? Can we keep increasing the test error by increasing the latent resolution?

**Limitations:**

1. As the paper stated, the trained model is limited to a specific observed category of shapes. The training for the operator requires a training dataset of high quality.

2. The proposed framework should be general for various kinds of PDEs, but only stable NS equations are tested.

3. The framework is for 3D PDEs, which could be modified for time-dependent ones.

---

> ### Author Rebuttal · Authors · 2023-08-09
>
> ## Response to Reviewer rW9d
>
> > **Q1:** Although the experiments have demonstrated the main ability of the model, the experiments are not complete enough to support all claims and novelties.
>
> **A1:** To further support the results, we add two more experiments that compare GINO with the solver and GNNs (MeshGraphNet). When comparing against the solver, we observe a four to five order of magnitude speed-up when using GINO. At a $3\%$ relative error, we find the speed-up from our model which includes drag in the loss to be $26,000 \times$. When comparing against GNN, we implemented the model following MeshGraphNet, which consists of an encoder, decoder, and processer. The model has a size of around 10M parameters, at a similar cost as other models. As shown in the updated experiments, the proposed model is faster than the solver while more accurate than other ML methods given a similar level of cost.
>
> > **Q2:** Since the model is a new combination of two existing methods GNO and FNO, the main contribution is the new usage of GNO. Then the main thing required to be demonstrated should be “GNO is more proper and has great encoding and decoding ability between irregular and regular grids”. Regarding this, some different encoding methods should be compared, such as GNN, kNN.
>
> **A2:** We agree that our main contribution is the combination of two existing methods GNO and FNO. Where GNO handles interaction on local irregular meshes and FNO learns global physics on the uniform latent grid. However, our goal is not to compare GNO against GNN or kNN in this paper. The main spirit of GNO is to design a graph structure with ball connectivity instead of nearest-neighbor connectivity, so the message passing is well-defined as the kernel integral on a ball. Indeed, modern graph neural networks as such GraphCast [1] also design the encoder and decoder with ball connectivity, following the definition of GNO. For a more comprehensive study, we do add a new experiment on GNN (MeshGraphNet) as discussed above.
>
> > **Q3:** Discretization invariance is said to be important, but no direct experiments could support this. Although Table 5 is relevant to this, the variables are not kept so it’s not a direct support. In other words, a GNN+FNO baseline should be added.
>
> **A3:** Discretization invariance means the model can be trained on one mesh discretization and used on another. For example, we have the super-resolution experiment where we train with sub-sampled (lower resolution meshes) and evaluate the model with the full mesh. If we design the encoder or decoder with the nearest-neighbor connectivity, then the neighbors (receptive field) will change along with the sampling rate, which makes it hard for the GNNs to generalize across different resolutions. Regarding the question, could the reviewer clarify which variables are not kept? We kept the same hyperparameters for all sampling rates.
>
> > **Q4:** What is the difference between your proposed GNO and Continuous Convolution (CtsConv) from “Lagrangian Fluid Simulation with Continuous Convolutions”? It seems there is no obvious difference based on equations. And one of the claimed contributions is a well-implemented GNO, which makes it much more efficient, but CtsConv is also well-implemented based on Open3D and hashing table.
>
> **A4:** Thanks for pointing out the paper. We will be happy to add a reference to it. CtsConv learns a continuous convolution for learning fluid mechanics which is based on linear interpolation.  CtsConv is an efficient local convolution layer that competes with GNNs and GNOs. Indeed our GNO implementation is similar to CtsConv. However, we consider 3D industry-level aerodynamic simulations. In the Ahmed-body dataset, the airflow has 7M particles and the surface mesh has 100k particles. As a comparison, the previous work only considers smaller-scale simulations with particles. For the large-scale problem, it's crucial to have an efficient FNO model to capture the global physical interaction, which is our main contribution.
>
> > **Q5:** What is the performance for much finer latent resolutions such as 128^3 and 256^3? Can we keep increasing the test error by increasing the latent resolution?
>
> **A5:** It's a very exciting direction to use much finer latent resolutions, which will likely further improve the performance as projected based on Table 3. However, 80^3 resolution already saturates 32GB Nvidia V100 GPUs. 128^3 and 256^3 resolution will take 4x and 32x more memory respectively. It's possible to push for higher resolution with the model parallel FNO [2], which we leave as future work.
>
> > **Q6:** As the paper stated, the trained model is limited to a specific observed category of shapes. The training for the operator requires a training dataset of high quality.
> The proposed framework should be general for various kinds of PDEs, but only stable NS equations are tested.
> The framework is for 3D PDEs, which could be modified for time-dependent ones.
>
> **A6:** Thanks for the comments, we would want to emphasize that 3D aerodynamics is a very one of the biggest industry problems. The time-averaged NS (RANS) is still the industry standard in car design.  We look forward to exploring time-dependent problems in future work.
>
> [1] Lam, Remi, et al. "GraphCast: Learning skillful medium-range global weather forecasting." arXiv preprint arXiv:2212.12794 (2022).
> [2] Grady, Thomas J., et al. "Model-parallel Fourier neural operators as learned surrogates for large-scale parametric PDEs." Computers & Geosciences (2023): 105402.

---

### Official Review · Reviewer_zqvK · 2023-07-08

**Soundness:** 3 good
**Presentation:** 2 fair
**Contribution:** 2 fair
**Rating:** 5
**Confidence:** 3

**Summary:**

This paper proposes a geometry-informed neural operator for arbitrary geometry, to facilitate the learning on the solution operator for large-scale 3D CFD simulation.

**Strengths:**

The proposed GINO model applies graph-kernel blocks for the encoder and decoder, for processing features in the latent uniform space, with the Fourier blocks running on the latent space to capture the global interaction. The proposed method provides significant runtime speedup.

**Weaknesses:**

For the experiment, the authors only perform the evaluation on the car category of ShapeNet dataset. It’ll be more persuasive to have the proposed model evaluate the other categories rather than only a single category.

**Questions:**

As the authors claim the significant speedup provided by the proposed method. Could the authors report the scale of the parameters of the proposed method, and the comparison to the other related methods?

**Limitations:**

As pointed out by the authors, this work is constrained to a specific category and limited to CFD with more complex shapes. It'll be with practical significance if the proposed work could tackle these limitations.

---

> ### Author Rebuttal · Authors · 2023-08-09
>
> ## Response to Reviewer zqvK
>
> > **Q1:** For the experiment, the authors only perform the evaluation on the car category of the ShapeNet dataset. It’ll be more persuasive to have the proposed model evaluate the other categories rather than only a single category.
>
> **A1:** Thanks to the reviewer for the comment. We want to first point out that we considered two datasets: one is the existing ShapeNet Car data; the other is the Ahmed body dataset we generated. 3D Aerodynamic simulations are extremely expensive. For the example of the Ahmed body dataset, each simulation takes 7-19 hours on 2 Nvidia v100 GPUs with 16 CPU cores.  It is extremely costly to generate a 3D dataset with multiple shapes. We have been searching through the literature for a very long but could not find another dataset resembling industry standard settings. If the reviewer would point out an existing dataset, we will be happy to try it.
>
> > **Q2:** As the authors claim the significant speedup provided by the proposed method. Could the authors report the scale of the parameters of the proposed method, and the comparison to the other related methods?
>
> **A2:** Thanks for the questions. For the larger Ahmed-body dataset, GINO uses O(100M) parameters which take around 5mins to train per each epoch. We match the FNO and UNet baselines to have a similar model size and training time.
> For smaller the Car-CFD dataset, we compared it against two other baselines: GeoFNO and GNO. The GeoFNO baseline is a 2D surface model. It's smaller and faster (10M parameters and 1 min training time), while its error is higher than the above 3D models. The GNO also has a smaller number of parameters (10M), but it has more edges that saturate the memory on an NVIDIA V100 GPU.
>
> To further support the results, we add two more experiments that compare GINO with the solver and GNNs (MeshGraphNet). When comparing against the solver, we observe a four to five order of magnitude speed-up when using GINO. At a $3\%$ relative error, we find the speed-up from our model which includes drag in the loss to be $26,000 \times$. When comparing against GNN, we implemented the model following MeshGraphNet, which consists of an encoder, decoder, and processer. The encoder and decoder have MLPs of channel size 256. The processer is 15 layers of message passing of edge blocks and node blocks. The model has a size of around 10M parameters. The details can be found in the general response. Note that the memory footprint of this model is the same as our 100M parameter model as they both saturate the memory of a single NVIDIA V100 GPU.
>
> As a conclusion, the proposed model is faster than the solver while more accurate than other ML methods given a similar level of cost.

---

### Author Rebuttal · Authors · 2023-08-09

# General response

We are grateful to the reviewers for their insightful feedback and constructive comments.
It is encouraging to note that part of the reviewers agree with the
- scalability of the proposed model to realistic 3D aerodynamic simulations,
- acknowledge the significant speed-up offered by our approach,
- and recognize our new dataset has been a valuable contribution.

However, the primary concerns lie in the comprehensiveness of the experiments. In response to these concerns and support our results, we add **two new experiments** that further compare our model with GNN methods and numerical solver.
- we added an experiment comparing GINO's performance with GNN. It shows GINO has a much smaller error rate compared to GNN (8.31% vs 13.88%).
- we perform a full cost-accuracy trade-off analysis. The result shows GINO is 26,000x faster at computing the drag coefficients.
- and we further expand the Ahmed body dataset from 500 instances to 800, making it the first and largest 3D RANS design dataset available for the community.

We sincerely hope these experiments make our study more comprehensive and clarify the concerns of the reviewers.

## 1. Comparison against GNNs

We added an experiment comparing GINO's performance with GNN. It shows GINO has a much smaller error rate compared to GNN (8.31% vs 13.88%).
We used the MeshGraphNet [1], as suggested by Reviewer 4. This model is a common GNN method for physical simulations. The MeshGraphNet has three main parts: an encoder, a decoder, and a processor. Both the encoder and decoder use the same setup, with a channel size of 256. The processor has 15 layers of information passing, using edge and node blocks. It also has a size of 256. In total, the model has about 10 million parameters.  The total number of edges is around 280k, which saturates a single NVIDIA V100 GPU with 32GB of memory. We set the learning rate to be 1e-4 with an exponential decay with a rate of 0.99985 similar to the original setup in [1].

The validation error of GNN for the pressure field is **13.88%**. The training curve is shown in the report. The error of GNN model is much higher than the GINO model (64x64x64 resolution, 100M params) of **8.31%**.  The smaller GINO (32x32x32 resolution, 10M params) can achieve 10.10% error rate. The GNN model is good at learning the local interaction with flexible meshes. However, it cannot easily learn the long-range global interactions. On the other hand, the GINO model has the advantages of both the graph and Fourier method. It has the flexibility of graphs as well as the efficiency of Fourier methods for capturing long-range dependencies.

[1] Pfaff, Tobias, Meire Fortunato, Alvaro Sanchez-Gonzalez, and Peter W. Battaglia. "Learning Mesh-Based Simulation with Graph Networks." Link: https://openreview.net/forum?id=roNqYL0_XP

## 2. Comparison against solver on Drag coefficients

To compare the performance of our model against the industry-standard OpenFOAM solver, we perform a full cost-accuracy trade-off analysis. The result shows GINO is **26,000x** faster at computing the drag coefficients. Figure (2, left) shows the cost-accuracy curve, measured in terms of inference time needed for a relative error in the drag coefficient for GINO and OpenFOAM.

The cost of GINO is computed as the time, averaged over the test set, needed to predict the drag coefficient by running the model. This time includes both data pre-processing (computing the SDF) as well as the model run-time and the drag calculation given the predicted fields. All models are run on a single NVIDIA V100 GPU. The cost for OpenFOAM is computed as described in the proceeding paragraph and is averaged over the test set. The solver is run on two NVIDIA V100 GPUs in parallel. We observe a four to five order of magnitude speed-up when using GINO. At a **3%** relative error, we find the speed-up from our model which includes drag in the loss to be 26,000 times. As we increase the size of the latent space, the cost of GINO grows, however, we observe a plateau in the drag error. This is common in machine learning models as the error from using finite data starts to dominate the approximation error. Furthermore, we use only the size of the latent space as a hyper-parameter, keeping the number of learnable parameters fixed. It is interesting to explore further how parametrically scaling the model impacts predictive power.

During data generation,
we keep track of the drag coefficient predicted by OpenFOAM after every iteration.
While the coefficient converges with more iterations, this convergence is not monotone and can often appear quite noisy. This makes computing the error from the raw data not possible. We therefore apply a box filter to the raw signal to compute a filtered version of the drag which acts as a smoother. We take as the reference drag, the drag at the last iteration of the filtered signal.
To compute the number of iterations it takes for the solver to predict a drag coefficient at a given relative error, we trace back the predictions from the filtered signal and return the first time at which this prediction incurs the given error with respect to the reference drag. An example of this methodology is shown in Figure (2, right). The errors for our GINO model are computed with respect to the true drag coefficient from the last iteration of the solver. This is because we take as ground truth the pressure and wall shear stress from this last iteration and train our model to predict them.

---

### Author Response · Authors · 2023-08-14
**Looking forward to your feedbacks**

We are happy to note that our responses have addressed the concerns of several reviewers. As the discussion deadline nears, we kindly urge the remaining reviewers to share their feedback. Your insights and comments are invaluable in enhancing the quality of our paper. Please inform us if there are any further concerns.

We believe that there's an agreement on the complexity and potential of 3D aerodynamics simulation. Our approach combined the flexibility of graphs with the efficacy of the spectral method, paving the way for scalable, complex 3D simulations. And we hope our dataset will lead to exciting future research in the community. If our rebuttal has partially resolved your concerns, we kindly request you to adjust the score accordingly. Your understanding and contributions are greatly appreciated.

---

### Decision · Program_Chairs · 2023-09-21

**Decision:**

Accept (poster)

**Comment:**

Reviewers for this paper agree that the proposed method seems solid and that the results are extremely impressive.  The main issue here seems to be that the method is undertested.  I am happy to suggest accepting this work with the understanding that the authors incorporate the new experiments/descriptions shared during the rebuttal phase.  Please also pay close attention to reviewer RPer's suggestions for ways to improve the exposition and avoid missing details.